# Uptake of malaria vaccine (RTS,S/AS01) among children aged 6–24 months: A cross-sectional survey conducted in the Western region of Cameroon, 1 year following the vaccine introduction

Jerome Ateudjieu[1,2,3], Merveille Claire Nana Djapou[2]*, Benjamin Kevin Bekoa Onana[2], Abdias Aron Tatsabong Tiomeni[2], Gretta Ludivine Okoumokath Mpande[2], Donald Kapso Nanguep[2], Collins Buh Nkum[2], Dora Winny Ateudjieu Kenfack[4,5], Anne Cecile Bissek[3]

1 Department of Public Health, Faculty of Medecine and Pharmaceutical Sciences, University of Dschang, Dschang, Cameroon, 2 Department of Health Research, Meilleur Accès aux Soins de Santé, Yaounde, Cameroon, 3 Division of Health Operations Research, Cameroon Ministry of Public health, Yaounde, Cameroon, 4 Department of Microbiology, faculty of Sciences, University of Yaounde 1, Yaounde, Cameroon, 5 Department of public health, Centre for Research on Emerging and Re-emerging Diseases CREMER, Yaounde, Cameroon

* merveille_claire@yahoo.com

## Abstract

Malaria vaccination using RTS,S/AS01 was introduced as part of Cameroon's EPI to prevent malaria morbidity and mortality among children aged 6–24 months. This study aims to assess the coverage, completeness, and timeliness of this vaccine and to explore parental perceptions of children's access to vaccination. This was a community-based, descriptive cross-sectional study targeting children aged 6–24 months and caregivers, selected by stratified random cluster sampling in Foumbot and Foumban health districts. Data were collected from caregivers by trained and supervised enumerators using a pretested questionnaire administered face-to-face. Malaria vaccine coverage, completeness, and timeliness were estimated with 95% confidence intervals. The contribution of caregivers' perception on the first dose malaria vaccination status was explored by estimating crude odds ratio (cOR) and adjusted odds ratio (aOR) estimated from mixed-effects logistic regression. Of the 55 targeted and reached clusters, 399 children were included in this study. Vaccination coverage of first, second, and third doses of the malaria vaccine was 31.20% (95%CI 30.38 - 32.02), 22.61% (95%CI 21.85 - 23.37), and 17.70% (95%CI 16.88 - 18.52), respectively. Among the children who received the first dose, 56.49% (95%CI 44.61 - 68.37) completed the 3-dose schedule. The timeliness of the three administered doses was between 50–57%. In multivariate analysis, parental perception remained significantly associated with dose 1 malaria administration (aOR 15.54 (95%CI 12.63 - 19.10) p < 0.001), dose 1 timeliness (aOR 2.36 (95%CI 1.52 - 3.65) p < 0.001) and

**Data availability statement:** The database has been Uploaded as supplementary information.

**Funding:** The funding for protocol development, field activities and manuscript drafting was supported by Meilleur Accès Aux Soins de Santé (M.A Sante), a Cameroon based NGO. The funder had no role in study design, data collection and analysis, decision to publish, or preparation of the manuscript.

**Competing interests:** The authors have declared that no competing interests exist.

dose 2 completeness (aOR 9.81 (95%CI 7.78 - 12.36) p < 0.001). More than a year after the malaria vaccine introduction in the health districts of Foumbot and Foumban, the performance indicators for this vaccination are below expectations for the vaccine to have a significant impact on reducing malaria morbidity in children. Communicating to positively influence caregivers' perceptions of malaria vaccination is expected to contribute to improving the situation.

## Background

Malaria remains a public health challenge worldwide, with the highest burden in Africa [1,2]. Several interventions have been prioritized to reduce the disease burden, including environmental interventions, the use of long-lasting mosquito nets, and prophylactic treatment for vulnerable groups such as pregnant women and children in malaria-endemic areas [3–6]. Although these interventions have contributed to a notable reduction in malaria morbidity and mortality, further efforts are required to achieve disease control [7]. In this context, vaccine development has led to the development of RTS,S/AS01, R21/Matrix-M, and PfSPZ vaccine. Among these, RTS,S/AS01 and R21/Matrix-M are currently recommended by the WHO [8–15].

Available literature documents the efficacy, safety, and benefits of malaria vaccination in various settings [7,11,16]. Evidence consolidated from pilot projects within the Expanded Program on Immunization (EPI) in Ghana, Kenya, and Malawi, along with subsequent WHO recommendations, supported the extension of malaria vaccination to other countries, including Cameroon [15–18]. This was followed by major reported challenges that included logistic difficulties, funding constraints, the complexity of administering the four-dose vaccine, and low caregivers' adherence in some contexts [19,20]. In Cameroon, RTS,S/AS01 was introduced in 2024 as part of the expanded national immunization program targeting children aged 6–24 months in 42 health districts with the highest malaria burden in children. Ten month after the introduction, administrative data reported a 44.8% third-dose coverage [21]. Exploring this coverage and caregivers' perceptions of malaria vaccination in households could generate evidence to guide interventions planning to reach the maximum number of targeted children.

## Methodology

### Ethics Statement

This study was conducted to generate evidence for the improvement of malaria vaccination in Cameroon. Its implementation involved household-based data collection from caregivers with the risk of increasing access to participants' confidential data and compromising participant autonomy. All participants were informed of the study objectives and procedures and were included only after providing a written consent form. Participants' confidentiality was protected by not collecting identifiable data, by password-protecting access to collected data, and by limiting access to the dataset to the team in charge of data management and analysis. The study protocol was

reviewed and approved by the West Region Ethics Review Committee for Human Health Research in Cameroon (reference number N°/580/29/05/2024/CE/CERSH-OU).

## Study design

This was a community-based descriptive cross-sectional study conducted from 5 to 13 May 2025, targeting children aged 6–24 months, selected by stratified random cluster sampling. Data were collected using a pretested questionnaire administered face-to-face to caregivers and a grid to review children's vaccination booklets. Malaria vaccine coverage, completeness, and timeliness were estimated from the collected data.

## Study setting

The study was conducted in the Foumban and Foumbot health districts in the West Region of Cameroon. At the time of the study, these were the only two districts in the region where the malaria vaccine was available under the EPI.

## Study population

Children aged 6 to 24 months residing in either the Foumbot or Foumban health district were eligible. Those with caregivers consenting to participate were included. Children without caregiver available to provide data on vaccination status, or those living in health areas with recent insecurity, were excluded.

## Sample size estimation

The minimum required sample size was estimated to be 392 mother-child couples. This calculation was based on an assumed a third-dose malaria vaccin coverage of 50%, a 95% confidence level, a 7% margin of error, and a cluster design effect of 2. Assuming approximately 7 children per cluster would be reached, the estimated sample was allocated across 55 clusters.

## Sampling process

The sampling process was stratified by the population size of the Foumban and Foumbot health districts. Six and three health areas, respectively, were randomly selected to be part of the study. For each health area, a list of communities and their respective population sizes was obtained. Clusters were allocated proportionally to communities' population size using a stratified systematic sampling approach. In targeted community participants of each cluster were selected by randomly choosing a starting household and a direction of recruitment. Households bordering the street were visited following the direction to identify and enroll study participants. Recruitment was considered completed in each cluster when approximately 7 children were included.

## Data collection tools and variables

Data were collected using a questionnaire administered face-to-face and a grid to review vaccination cards. These tools were adapted from an existing questionnaire that was originally developed to assess documented and undocumented EPI vaccination coverage, timeliness, and completeness [22,23]. The main data collected included the sociodemographic characteristics of the mothers and children, the child's malaria vaccination status, and the caregivers' exposure to malaria vaccination information, and their perception of the malaria vaccination benefits and safety.

## Data collection process

Data collection was performed by teams of three members including two trained enumerators and one trained supervisor. The teams were introduced by the supervisors in each household after obtaining the permission from the household head.

If an eligible child (children aged 6–24 months) was present, informed consent was obtained from the caregiver. Data were then collected from the cargiver using a face-to-face administered questionnaire. If the child had vaccination booklet, a data extraction form was used to collect data on the vaccination status. Data were collected electronically using the Kobocollect mobile interface, uploaded online after verification by supervisors, and downloaded daily by the data management team for review. Queries were sent back to enumerators for corrections when necessary.

## Data analysis

We estimated the documented vaccine coverage, timeliness, and completeness using numerators and denominators as presented in Table 1. The analysis was conducted using sampling weights. The individual weights corresponded to the ratio between the total population of each health area and the number of children actually surveyed in that area. Weighting allowed reconstruction of the target population distribution without further normalization. We explored the association between caregivers' perception of the malaria vaccine on first malaria documented dose coverage and second dose malaria vaccine completeness using a case-control approach, estimating crude and adjusted odds ratios. Adjustment was done using mixed-effects logistic regression, with confounders identified from the comparison of key characteristics between cases and controls. Controls were children with documented vaccination, and cases were children with no documented vaccination. A positive perception was defined as reporting that "the vaccine is safe" OR "the vaccine helps prevent malaria" NOR any negative opinion. Conversely, participants who reported that "the vaccine is not effective" OR "the vaccine can kill" OR "the vaccine causes adverse effects" NOR any positive opinion were classified as having a negative

**Table 1. Definition of denominators and numerators for the estimate of the documented vaccine coverage, timeliness, completeness of the malaria vaccines uptake.**

| Estimated indicator | Numerator | Denominator |
|---|---|---|
| Documented coverage first dose malaria vaccine dose | Number of children documented to have received the first malaria vaccine dose | Number of children aged 7–24 months |
| Documented coverage second dose malaria vaccine dose | Number of children documented to have received the second malaria vaccine dose | Number of children aged 8–24 months |
| Documented coverage third dose malaria vaccine dose | Number of children documented to have received the third malaria vaccine dose | Number of children aged 12–24 months of age |
| Timeliness first malaria dose | Number of children documented to have received the first malaria vaccine dose before the end of 7th months of age | Number of children vaccinated with first dose malaria vaccine |
| Timeliness second malaria dose | Number of children documented to have received the second malaria vaccine dose before the end of 8th months of age | Number of children vaccinated with second dose malaria vaccine |
| Timeliness third malaria dose | Number of children documented to have received the third malaria vaccine dose before 11th months of age | Number of children vaccinated with third dose malaria vaccine |
| Two doses completeness among those exposed to 01 dose | Number of children aged 8–24 months (documented) that have received two doses of malaria vaccine | Number of children 8–24 months (documented) that received the first dose of malaria vaccine |
| Two doses completeness among children reached | Number of children aged 8–24 months (documented) that have received two doses of malaria vaccine | Number of children aged 8–24 months |
| Completeness third dose | Number of children aged 12–24 months (documented) that have received three doses of malaria vaccine | Number of children aged 12–24 months (documented) that received the first dose of malaria vaccine |

**PLOS** **Global Public Health**

perception. Individuals who reported that "the vaccine is safe" OR "the vaccine helps prevent malaria" AND "the vaccine is not effective" OR "the vaccine can kill" OR "the vaccine causes adverse effects" were classified as having a mixed perception and were excluded from the analysis. Data were analyzed using SPSS version 24.0 and R version 4.4.2. Estimates were reported with 95% confidence intervals.

## Results

### Survey coverage

We reached the 55 targeted clusters and 399 participants, all of whom consented to participate, with average cluster coverage of 93.41% (95% CI 91.32-95.50).

### Socio-demographic characteristics of caregivers and children

The mean age of the caregivers and children was $27.38 \pm 7.45$ years and $14.24 \pm 5.41$ months, respectively. Socio-demographic characteristics of the study population are presented in Table 2. The majority of caregivers (97.1%) were female; 11.5% were adolescents, and 37.5% had a low level of education. Table 3 presents the socio-demographic characteristics of the children.

### Vaccine coverage, completeness and timeliness

Of the 399 children reached, 208 (51.9%) were reported by caregivers to have received at least 1 dose of malaria vaccine, while 121 (29.8%) were documented to have received at least 1 dose of malaria vaccine.

Children's vaccination coverage, timeliness, and completeness are presented in Table 4. Among included children, 48 (17.70%) were documented as having received the third dose of malaria vaccine. Of the 119 children who received the first dose, 46 (37.98%) completed the three-dose schedule. Timeliness rates for the three doses ranged from 50.34% to 56.66%.

### Caregivers' perception of the malaria vaccine

Caregivers' perception regarding the malaria vaccine is presented in Table 5. Of the 399 included caregivers, 325 (81.8%) had positive perceptions of vaccination.

### Comparison of the distribution of socio-demographic characteristics between vaccinated and unvaccinated children

Table 6 presents the distribution of sociodemographic characteristics between vaccinated and unvaccinated children. There was a significant difference in sociodemographic characteristics between children who received the first dose of the malaria vaccine and those who did not. However, no significant differences were observed for caregiver age ≤ 19 years, occupation (farmer or housekeeper), or religion (Muslim) when comparing children who received the first dose on time with those who did not, or for second-dose completeness.

### Association between caregivers' perception and vaccination status of children

Table 7 presents the contribution of malaria vaccine perception to children's vaccination status. Children whose caregivers had a positive perception of the malaria vaccine were more likely to receive the first dose, to receive it on time, and to complete two doses.

## Discussion

To our knowledge, this is the first community-based survey exploring the coverage, completeness, and timeliness of malaria vaccination since its introduction in Cameroon. Our findings indicate documented vaccine coverage was 31.20%

**Table 2. Socio-demographic characteristics of caregivers.**

| Variable | Modalities | Frequency | Percentage (%) |
|---|---|---|---|
| **Sex** | Male | 12 | 2.9 |
| | Female | 387 | 97.1 |
| **Age (years)** | ≤ 19 | 45 | 11.5 |
| | 20–49 | 347 | 87 |
| | 50 and above | 6 | 1.5 |
| **Caregiver's relationship with the child** | Mother | 373 | 93.8 |
| | Father | 11 | 2.7 |
| | Grandmother | 7 | 1.7 |
| | Older brother or sister | 5 | 1.1 |
| | Member of the parents' family | 3 | 0.8 |
| **Level of education** | No school | 19 | 4.5 |
| | Primary | 129 | 33.0 |
| | Combined low level of education (No school and primary) | 148 | 37.5 |
| | Secondary | 242 | 60.2 |
| | University | 9 | 2.3 |
| **Occupation** | Housekeeper | 199 | 50.0 |
| | Farmer | 73 | 17.4 |
| | Civil servant | 12 | 2.1 |
| | Entrepreneur | 10 | 2.6 |
| | Shopkeeper | 24 | 6.1 |
| | Dressmaker | 47 | 11.8 |
| | Other | 34 | 10.1 |
| **Religion** | Christian | 79 | 19.3 |
| | Muslim | 319 | 80.4 |
| | Animist | 1 | 0.3 |
| **Floor type of household inhabited by the caregiver** | Tiles | 57 | 14.6 |
| | Cemented | 208 | 50.1 |
| | Carpet | 5 | 1.3 |
| | Clay | 139 | 34.0 |
| **Average number of people in the household of the caregiver** | 6.06±2.71    95% CI = (6.013; 6.107) | | |

**Table 3. Socio-demographic characteristics of children.**

| Variables | Modalities | Effectives | Percentage |
|---|---|---|---|
| **Sex** | **Male** | 190 | 47.6 |
| | **Female** | 209 | 52.4 |
| **Age (months)** | **6-11** | 135 | 34.1 |
| | **12-24** | 264 | 65.9 |

**Table 4. Malaria vaccine coverage, completeness and timeliness.**

| | Frequency | Percentage | 95% CI |
|---|---|---|---|
| **Documented coverage per dose** | | | |
| Dose 1 (n = 372 children aged 7–24) | 119 | 31.20 | 30.38 - 32.02 |
| Dose 2 (n = 348 children aged 8–24) | 80 | 22.61 | 21.85 – 23.37 |
| Dose 3 (n = 264 children aged 12–24 months) | 48 | 17.70 | 16.88 – 18.52 |
| **Completeness** | | | |
| Two doses completeness among those exposed to 01 dose (n = 102 children aged 8–24) | 77 | 74.76 | 66.33 – 83.19 |
| Two doses completeness among children reached (n = 348 children aged 8–24) | 77 | 21.65 | 16.72 – 25.28 |
| Three doses completeness among those exposed to 01 dose (n = 67 children aged 12–24) | 39 | 56.49 | 44.61 – 68.37 |
| **Timeliness** | | | |
| 1st dose (n = 119) | 66 | 55.24 | 53.6 – 56.8 |
| 2nd dose (n = 80) | 40 | 50.35 | 48.4 – 52.3 |
| 3rd dose (n = 48) | 27 | 56.66 | 48.1– 53.2 |

**Table 5. Caregiver perceptions of malaria vaccination.**

| Variable | | N (%) |
|---|---|---|
| **Positive perception** | Vaccine is safe | 159 (39.1%) |
| | Vaccine helps to prevent malaria | 284 (71.7%) |
| | Vaccine is safe **AND** vaccine helps to prevent malaria | 114 (28.2%) |
| | Vaccine is safe **OR** vaccine helps to prevent malaria **NOR** Negative opinion | 325(81.8%) |
| **Negative perception** | The Vaccine is not efficient | 9 (2%) |
| | The Vaccine can kill | 8 (1.9%) |
| | The vaccine induces adverse effects | 13 (3.2%) |
| | The Vaccine is not efficient **AND** The Vaccine can kill **AND** The vaccine induces adverse effects | 0(0) |
| | The Vaccine is not efficient **OR** The Vaccine can kill **OR** The vaccine induces adverse effects **NOR** Positive opinion | 23(5.7%) |
| **Mixed opinion** | Vaccine helps to prevent malaria **AND** The Vaccine is not efficient | 3(0.7%) |
| | Vaccine helps to prevent malaria **AND** The vaccine induces adverse effects | 1(0.1) |

for dose 1, 22.61% for dose 2, and 17.70% for dose 3. The completeness of the second and third doses was 74.76% and 56.49%, respectively, and the timeliness of the first, second, and third doses was 55.24%, 50.35%, and 56.66%, respectively. Children's exposure to caregivers with a positive perception of malaria vaccination were more likely to receive the first dose, receive it on time, and complete two doses.

Vaccination coverage reflects the proportion of the target population that has benefited from the intervention. It is also an indicator of vaccine demand and of the effectiveness of strategies implemented to reach the target vaccinated population. This study revealed that less than one-third of eligible children were documented as having received the first dose of the malaria vaccine. The second and third dose coverage of the malaria vaccination dropped to less than one-quarter and to less than one-fifth, respectively. In contrast, a study analyzing administrative data one month after the introduction of malaria vaccination in Cameroon by the national EPI in the 42 targeted health districts reported a first-dose malaria vaccine coverage relatively higher (37% of eligible children) than that observed in our study [24]. Furthermore, fourteen months after the introduction of the malaria vaccine in Cameroon, and one month before the data collection of this study,

**Table 6. Sociodemographic characteristics distribution between selected vaccination status of children.**

| Variables | Modalities | Dose 1 malaria administration | | p-value |
|---|---|---|---|---|
| | | Yes | No | |
| Cemented floor | Yes | 74(37%) | 126(63%) | **<0.001** |
| Caregiver being farmer | Yes | 8(11.4%) | 62(88.6%) | **<0.001** |
| Caregiver being Muslim | Yes | 92(28.8%) | 227(71.2%) | **<0.001** |
| Caregiver had Low level of education | Yes | 32(21.6%) | 116(78.4%) | **<0.001** |
| Caregiver being Mother | Yes | 117(17.2%) | 256(82.8%) | **<0.001** |
| Caregiver being female | Yes | 119(30.7%) | 268(69.3%) | **<0.001** |
| Caregiver being Housekeeper | Yes | 61(30.7%) | 138(69.3%) | **<0.001** |
| Caregiver being ≤ 19 years | Yes | 15(33.3%) | 30(66.7%) | **<0.001** |
| | | **Dose 1 malaria vaccine Timeliness** | | |
| | | Yes | No | |
| Cemented floor | Yes | 38(51.4%) | 36(48.6) | **<0.001** |
| Caregiver being farmer | Yes | 4(49.4%) | 4(50.6%) | 0.057 |
| Caregiver being Muslim | Yes | 51(55%) | 41(45%) | 0.530 |
| Caregiver had Low level of education | Yes | 19(60.4%) | 13(39.6%) | **<0.001** |
| Caregiver being Mother | Yes | 64(54.5%) | 53(45.5%) | **<0.001** |
| Caregiver being Housekeeper | Yes | 34(55.5%) | 27(44.5%) | 0.742 |
| Caregiver being ≤ 19 years | Yes | 7 (44.8%) | 8(55.2%) | **<0.001** |
| | | **Dose 2 malaria vaccine completeness** | | |
| | | Yes | No | |
| Cemented floor | Yes | 47 (26.1) | 128 (73.9) | **0.001** |
| Caregiver being farmer | Yes | 4 (6.3) | 56 (93.7) | **<0.001** |
| Caregiver being Muslim | Yes | 64 (22.3) | 214 (77.7) | **<0.001** |
| Caregiver had Low level of education | Yes | 19 (15.2) | 105 (84.8) | **<0.001** |
| Caregiver being Female | Yes | 77 (22.3) | 260 (79.7) | **<0.001** |
| Caregiver being Mother | Yes | 75 (22.7) | 248 (77.3) | **<0.001** |
| Caregiver being Housekeeper | Yes | 39 (23) | 127 (77) | **0.001** |
| Caregiver being ≤ 19 | Yes | 7 (20.3) | 27 (79.7) | 0.256 |

**Table 7. Bivariate association between malaria vaccine perception and vaccination status of children.**

| Variables | Modalities | OR [95% CI] | P-Value | AOR [95% CI] | P-value |
|---|---|---|---|---|---|
| **Dose 1 malaria administration** | | | | | |
| Positive perception of malaria vaccine | No (reference) | | | | |
| | Yes | 12.66 [10.40-15.40] | **<0.001** | 15.54[12.63-19.10] | **<0.001** |
| **Dose 1 malaria vaccine timeliness** | | | | | |
| Positive perception of malaria vaccine | No (reference) | | | | |
| | Yes | 2.32[1.51-3.57] | **<0.001** | 2.36[1.52-3.65] | **<0.001** |
| **Dose 2 malaria vaccine completness** | | | | | |
| Positive perception of malaria vaccine | No (reference) | | | | |
| | Yes | 8.42[6.75-10.50] | **<0.001** | 9.81[7.78-12.36] | **<0.001** |

*cOR: Crude odds ratio; *AOR: Adjusted Odds ratio; *significant association cut-off: p<0.05

the cumulative three-dose malaria vaccination coverage from January to April 2025, as reported by the national Expanded Program on Immunization, was 64.3%, 54.1%, and 62.3% at the national level and 53.9%, 40.6%, and 42.2% in the two health districts targeted by our study for the first, second, and third doses of the malaria vaccine, respectively [25]. The discrepancies between administrative data and this study's findings remain to be fully understood. They may be explained by several factors: children accessed during vaccination sessions are not rigorously selected from communities within the target health districts; our study estimated only documented vaccination; and there is a potential for error in estimating either the numerator or denominator used for administrative vaccination coverage.

The relatively higher malaria vaccination coverage observed after the vaccine introduction in other countries can be explained by better communication prior to the vaccination introduction, resulting in better adherence of the caregivers and/or by vaccination strategies [16,26]. The coverage observed in our study was below the expectation of the national Cameroon EPI [27]. Assuming that the malaria vaccine introduced in Cameroon (RTS,S/AS01) maintains its efficacy as reported in clinical trials, the three-dose vaccination coverage reported in our study predicts that nearly one in twenty children in the study districts will be fully protected by their first birthday. This underlines the urgent need, on one hand, to investigate the determinants of poor performance of malaria vaccination and, on the other hand, to implement documented interventions that have been shown to contribute to improve EPI coverage in similar contexts [23].

The vaccination completeness indicator is important for monitoring performance immunization programs [28]. It also provides information on the health system's capacity to ensure optimal protection for vaccinated children, as vaccine effectiveness depends on full compliance with the recommended vaccination schedule for each given vaccine. Malaria vaccination completeness is necessary to maximize individual immunity and also to significantly reduce malaria morbidity and mortality in children [11]. In this study, half of the children who started the malaria vaccine completed the third dose. The observed high dropout rate in our study aligns with varying with varying levels of perceived importance of the malaria vaccine and other EPI vaccines as documented from studies conducted in similar and other contexts [29]. The present study was not designed to understand factors contributing to the malaria vaccination dropout rate, evidence from studies of other vaccines suggests that negative maternal attitudes toward vaccination, home birth, and limited geographic access may increase the risk of incomplete vaccination [30]. Further studies should explore the contribution of these factors to malaria vaccination completeness and other factors, such as the scheduling of vaccination sessions and the occurrence of adverse events after vaccination, which have been identified as inducing vaccination hesitancy [31]. The benefit of other caregivers' reminders using SMS or tracking of children's vaccination status in households by community volunteers to catch up on missed doses, shown to be beneficial for other vaccines, should be explored for their contribution to improving malaria vaccination completeness [23,31].

The timing and recommended interval between vaccine doses are based on the immune response and child development, as assessed through clinical trials and age-specific disease risks [32,33]. To our knowledge, no previous studies have assessed the timeliness of malaria vaccination in Cameroon. Like all EPI vaccines, malaria vaccination has an administration schedule, it lacks an operational definition of vaccination timeliness. In this study, we considered a dose timely if administered within one month of the recommended age. Under this definition, approximately half of the children received each dose on time. Comparing this trend with the timeliness of other EPI vaccines administered in the same context a few years earlier shows a higher timeliness rate for the BCG vaccine, which is administered at birth, and a timeliness rate consistent with our study for the DPT-Hi + Hb 3 vaccine, i.e., nearly half of children receiving the third dose of this vaccine within one month of the age recommended by the EPI [29]. This consistency likely reflects that nearly half of vaccinated children received the vaccine later than recommended. This delay is related to the supply or demand of all Expanded Program on Immunization (EPI) vaccines, not just the malaria vaccine, and must be addressed programmatically. Follow-up studies of children in high-risk areas show an increase in malaria incidence with time after each malaria vaccine dose, suggesting that some benefit of the malaria vaccine is lost when the vaccine administration schedule is not respected [11]. Making available an operational definition of the timeliness of each vaccine dose, conducting studies

to understand low vaccine timeliness, and identifying interventions to improve it should guide the Expanded Program on Immunization to improve malaria vaccine timeliness.

Caregivers' perception of children's vaccination is expected to influence its demand for the benefit of children. In our study, a positive perception of the malaria vaccine was significantly associated with increase children's access to the first malaria dose as well as the timeliness of the dose and the completeness of the second dose. To our knowledge, no published study has investigated the contribution of caregiver perception to malaria vaccines and children's access to this vaccine. In Malawi, evidence in line with our finding was documented by showing the contribution of caregivers' exposure to negative rumors about the malaria vaccine and a significant reduction of children's likelihood of full uptake of the malaria vaccine [34]. This finding extend beyond malaria vaccine; a systematic review of general childhood vaccination revealed that negative perceptions of vaccines were significantly associated with non-use of vaccination [35]. Therefore, generating evidence on innovative interventions to improve caregivers' perception of the malaria vaccine is expected to guide decision makers in taking needed action to improve malaria vaccine uptake in children.

The results of this study should be used taking into consideration the following limitations. Malaria vaccination coverage, timeliness, and completeness were estimated based on documented vaccination status. This could underestimate access to vaccination, as a given proportion of vaccinated children may be vaccinated without documentation [36].

## Conclusion

In conclusion, the coverage, completeness, and timeliness of malaria vaccination are below the targets needed to achieve optimal vaccination benefits. Children's exposure to caregivers with a positive perception of malaria vaccination increased their chance of receiving the first dose of the malaria vaccine, of receiving this dose on time, and of completing two doses of the malaria vaccine. We recommend that the competent health authorities in the targeted health districts implement communication initiatives aimed at improving caregivers' perceptions on malaria vaccination; that scientists test innovative interventions to improve caregivers' perceptions on malaria vaccination; and that other determinants of children's access to malaria vaccination be explored to guide decision-making on promoting children's access to the malaria vaccine.

## Supporting information

**S1 Data. Database.**
(SAV)

## Acknowledgments

We thank the health authorities of the West Region of Cameroon for facilitating and accommodating the implementation of the project. We also thank all enumerators and supervisors for their contribution to the data collection.

## Author contributions

**Conceptualization:** Jerome Ateudjieu.

**Data curation:** Merveille Claire Nana Djapou, Gretta Ludivine Okoumokath Mpande.

**Formal analysis:** Benjamin Kevin Bekoa Onana.

**Funding acquisition:** Jerome Ateudjieu.

**Investigation:** Jerome Ateudjieu, Abdias Aron Tatsabong Tiomeni, Donald Kapso Nanguep.

**Methodology:** Jerome Ateudjieu.

**Project administration:** Jerome Ateudjieu.

**Supervision:** Jerome Ateudjieu, Anne Cecile Bissek.

**Validation:** Jerome Ateudjieu.

**Writing – original draft:** Jerome Ateudjieu, Merveille Claire Nana Djapou, Abdias Aron Tatsabong Tiomeni, Gretta Ludivine Okoumokath Mpande, Donald Kapso Nanguep.

**Writing – review & editing:** Jerome Ateudjieu, Merveille Claire Nana Djapou, AAbdias Aron Tatsabong Tiomeni, Gretta Ludivine Okoumokath Mpande, Donald Kapso Nanguep, Collins Buh Nkum, Dora Winny Ateudjieu Kenfack.

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
