## [Decision Letter · Decision Letter 0]

23 Nov 2025

PGPH-D-25-02818

Malaria vaccination coverage, completeness, and timeliness among children aged 6–24 months: a cross-sectional survey conducted in the Western region of Cameroon, 1 year following the vaccine introduction

Dear Dr. NANA DJAPOU,

Thank you for submitting your manuscript to PLOS Global Public Health. After careful consideration, we feel that it has merit but does not fully meet PLOS Global Public Health’s publication criteria as it currently stands. Therefore, we invite you to submit a revised version of the manuscript that addresses the points raised during the review process.

We look forward to receiving your revised manuscript.

Kind regards,

André Machado Siqueira, M.D., MSc, Ph.D

Academic Editor

Journal Requirements:

1. Please amend your detailed online Financial Disclosure statement. This is published with the article. It must therefore be completed in full sentences and contain the exact wording you wish to be published.

a) State the initials, alongside each funding source, of each author to receive each grant. For example: “This work was supported by the National Institutes of Health (####### to AM; ###### to CJ) and the National Science Foundation (###### to AM).”

For more information, please go to our submission guidelines:

https://journals.plos.org/globalpublichealth/s/submission-guidelines#loc-financial-disclosure-statement

2. Please ensure that the funders and grant numbers match between the Financial Disclosure field and the Funding Information tab in your submission form. Note that the funders must be provided in the same order in both places as well.

3. Please update your online Competing Interests statement. If you have no competing interests to declare, please state: “The authors have declared that no competing interests exist.”

4. In the online submission form, you indicated that “The database is available and accessible from the corresponding author.”.

a) In a public repository,

b) Within the manuscript itself, or

c) Uploaded as supplementary information.

For further assistance, you may go to: http://journals.plos.org/globalpublichealth/s/data-availability

5. Please provide separate main figure files in .tif or .eps format only and ensure that all files are under our size limit of 10MB.

6. We do not publish any copyright or trademark symbols that usually accompany proprietary names, eg (R), (C), or TM (e.g. next to drug or reagent names). Please remove all instances of trademark/copyright symbols throughout the text, including ™ on page 18.

7. Some material included in your submission may be copyrighted. According to PLOS’s copyright policy, authors who use figures or other material (e.g., graphics, clipart, maps) from another author or copyright holder must demonstrate or obtain permission to publish this material under the Creative Commons Attribution 4.0 International (CC BY 4.0) License used by PLOS journals. Please closely review the details of PLOS’s copyright requirements here: PLOS Licenses and Copyright. If you need to request permissions from a copyright holder, you may use PLOS's Copyright Content Permission form.

Potential Copyright Issues:

Figure 1: please (a) provide a direct link to the base layer of the map (i.e., the country or region border shape) and ensure this is also included in the figure legend; and (b) provide a link to the terms of use / license information for the base layer image or shapefile. We cannot publish proprietary or copyrighted maps (e.g. Google Maps, Mapquest) and the terms of use for your map base layer must be compatible with our CC-BY 4.0 license.

Additional Editor Comments (if provided):

Reviewers' comments:

Reviewer's Responses to Questions

**Comments to the Author**

1. Does this manuscript meet PLOS Global Public Health’s publication criteria?

Reviewer #1: No

Reviewer #2: Yes

2. Has the statistical analysis been performed appropriately and rigorously?

Reviewer #1: Yes

Reviewer #2: Yes

3. Have the authors made all data underlying the findings in their manuscript fully available (please refer to the Data Availability Statement at the start of the manuscript PDF file)?

Reviewer #1: Yes

Reviewer #2: Yes

4. Is the manuscript presented in an intelligible fashion and written in standard English?

Reviewer #1: Yes

Reviewer #2: Yes

Reviewer #1: Malaria remains a major public health challenge despite multiple interventions that have reduced its prevalence and impact globally and in Cameroon . The introduction of new malaria vaccines is a welcome addition to the malaria control tool kit. The authors have reported on a survey in which they determined the coverage, completeness and timeliness of the distribution of a malaria vaccine in the western region of Cameroon. They furthermore assed the impact of factors such as knowledge and perception of the malaria vaccine on the studied parameters. They found extremely high drop out rates and negative perceptions of the malaria vaccine. This is surprising since the expanded program on immunization (EPI) vaccines' uptake in Cameroon is generally high. The interpretation of their results would have been more convincing if they had a controlled group taking a comparator vaccine, or the same group administered one of the EPI vaccines for comparison. As it stands their study is seriously flawed by the lack of appropriate controls which makes interpretation difficult, if not impossible.

Furthermore the authors did not attempt to determine the efficacy and impact of the vaccine on the prevalence and morbidity of malaria during the study period. As it stands this looks like a progress report on an interesting and relevant study which which needs to be properly controlled in order to yield credible results. To improve on the manuscript, the authors should include the name of the malaria vaccine studied i.e., RTS,S in the title (which should be shortened) and the Abstract. Based on what has been said above, I recommend that the manuscript be returned to the authors for a major revision.

Reviewer #2: The manuscript is suitable for publication after minor revisions. Its novelty, rigorous methodology, and relevant findings make it valuable for this journal. However, the following issues need to be addressed in the manuscript;

1. The manuscript requires proof editing for grammar.

2. Provide clearer definition for "positve perception" and specify how variables were dichotomised.

3. Explain how weighing was applied in the analysis.

4. Clarify operational definitions of timeliness since these are non-standard for malaria vaccines.

5. The data collection process section should be improved.

**Do you want your identity to be public for this peer review?** For information about this choice, including consent withdrawal, please see our Privacy Policy

Reviewer #1: No

Reviewer #2: No

---

## [Decision Letter · Decision Letter 1]

28 Jan 2026

Uptake of malaria vaccine ( RTS,S/AS01) among children aged 6–24 months: a cross-sectional survey conducted in the Western region of Cameroon, 1 year following the vaccine introduction

PGPH-D-25-02818R1

Dear Mrs NANA DJAPOU,

We are pleased to inform you that your manuscript 'Uptake of malaria vaccine ( RTS,S/AS01) among children aged 6–24 months: a cross-sectional survey conducted in the Western region of Cameroon, 1 year following the vaccine introduction' has been provisionally accepted for publication in PLOS Global Public Health.

Best regards,

André Machado Siqueira, M.D., MSc, Ph.D

Academic Editor

Reviewer Comments (if any, and for reference):

Reviewer's Responses to Questions

**Comments to the Author**

Reviewer #1: All comments have been addressed

Reviewer #2: All comments have been addressed

publication criteria?

Reviewer #1: Yes

Reviewer #2: Yes

3. Has the statistical analysis been performed appropriately and rigorously?

Reviewer #1: Yes

Reviewer #2: Yes

4. Have the authors made all data underlying the findings in their manuscript fully available (please refer to the Data Availability Statement at the start of the manuscript PDF file)?

Reviewer #1: Yes

Reviewer #2: Yes

5. Is the manuscript presented in an intelligible fashion and written in standard English?

Reviewer #1: Yes

Reviewer #2: Yes

Reviewer #1: The authors have satisfactorily addressed my main concern which was about the low RTS,S malaria vaccine uptake in the study area. The revised manuscript is recommended for publication in PLOS Public Health journal.

Reviewer #2: Welldone for the attempt made to answer all the reviewers comments raised.

**Do you want your identity to be public for this peer review?** For information about this choice, including consent withdrawal, please see our Privacy Policy

Reviewer #1: No

Reviewer #2: No
